# Microwave NDT of Smart Composite Structures with Embedded Antennas

**DOI:** 10.3390/s23063200

**Published:** 2023-03-17

**Authors:** Mohammed Saif ur Rahman, Omar Samir Hassan, Ademola Akeem Mustapha, Mohamed A. Abou-Khousa, Wesley James Cantwell

**Affiliations:** 1Electrical Engineering and Computer Science Department, Khalifa University of Science and Technology, Abu Dhabi P.O. Box 127788, United Arab Emirates; 2Department of Aerospace Engineering, Khalifa University of Science and Technology, Abu Dhabi P.O. Box 127788, United Arab Emirates

**Keywords:** antenna, composites, conformal load-bearing antenna structures, microwaves, non-destructive testing, smart structures

## Abstract

The integration of antennas in composite structures is gaining popularity with advances in wireless communications and the ever-increasing demands for efficient smart structures. Efforts are ongoing to ensure that antenna-embedded composite structures are robust and resilient to inevitable impacts, loading and other external factors that threaten the structural integrity of these structures. Undoubtedly, the in situ inspection of such structures to identify anomalies and predict failures is required. In this paper, the microwave non-destructive testing (NDT) of antenna-embedded composite structures is introduced for the first time. The objective is accomplished using a planar resonator probe operating in the UHF frequency range (~525 MHz). High-resolution images of a C-band patch antenna fabricated on an aramid paper-based honeycomb substrate and covered with a glass fiber reinforced polymer (GFRP) sheet are presented. The imaging prowess of microwave NDT and its distinct advantages in inspecting such structures are highlighted. The qualitative as well as the quantitative evaluation of the images produced by the planar resonator probe and a conventional K-band rectangular aperture probe are included. Overall, the potential utility of microwave NDT for the inspection of smart structures is demonstrated.

## 1. Introduction

Antennas are extensively used in aerospace and space platforms for various applications encompassing radars, communication, positioning, etc. The concept of embedding RF antennas onto load-bearing aircraft skins was introduced a couple of decades ago for aerospace and defense applications and since then there has been keen interest concerning the integration of antennas in composite structures to improve structural efficiency and create smart structures. The term conformal load-bearing antenna structures (CLAS) is often used to describe such structures [1,2,3]. Over the years, several developments have been proposed in this regard such as surface antenna structures, composite smart structures and 3D integrated microstrip antenna structures to further enhance structural and electrical efficiency [4,5,6,7,8]. Recently, a frequency-modulated continuous wave (FMCW) radar front-end was embedded inside an aerospace composite [9]. The manufacturing process for highly integrated embedded antennas is very complex and can lead to irregularities. Furthermore, such structures are always prone to impact damage, loading, vibrations and other factors that could threaten the structural integrity of the component. This could inevitably affect the performance of the structure and may lead to the complete failure of the part. Therefore, ensuring the quality of the build and integrity of the antenna before putting it into service is essential. Also, given the nature of the composite antenna itself (a woven mesh) it could be damaged whilst in service, thereby requiring subsequent localization and repair. This can be accomplished through non-destructive testing (NDT), which is well established and renowned. The study of antenna performance in the presence of impact damage, delamination and dynamic loading has been undertaken by various researchers in the past [10,11,12]. The metric to gauge the structural integrity of the antenna-embedded composite was largely based on the antenna measurement characteristics, such as radiation pattern, return loss, etc., all of which require a special infrastructure, equipment and the isolation of the structure during testing. The in situ inspection of antenna-embedded composite structures has not been explored in the literature, to the best of the authors’ knowledge. Distinct advantages are associated with in situ monitoring, including the prediction of any impending failure in the structure and a relatively short inspection time frame. Within the realm of NDT, there are numerous techniques that can be employed to perform the structural integrity assessment of structures under test (SUT). However, with regard to the inspection of composite structures, several NDT modalities have limited applicability as highlighted multiple times in the past [13,14,15,16,17]. X-ray computed tomography (X-ray CT) is the gold NDT standard in this application for both quality check/control and inspection. X-ray CT is expensive and the machine can only accommodate patches (not an entire functional surface), which increases the need for an alternative method.

As a result of their inherent advantages, microwave and millimeter waves have been successfully employed in the past to non-destructively test composites of various types including fiberglass, honeycomb, radome, etc. using waveguide apertures and planar resonant probes [17,18,19,20,21,22,23,24,25,26,27]. However, the microwave inspection of antennas embedded in composites is new and has not been investigated before. In this paper, we introduce microwave non-destructive testing as an alternative method for the inspection of CLAS for the first time. To illustrate the utility of microwaves for antenna-embedded structures, a microwave planar resonator probe operating in the ultrahigh frequency (UHF) range is utilized. Among various microwave probes, the planar resonator probe, being sensitive to current distributions, is deemed advantageous in this application [28,29,30].

This paper is organized in the following manner. The detection concept is briefly explained in Section 2 in addition to simulation studies and the analysis of the antenna-embedded composite structure. Imaging results of the antenna-embedded composite structure are subsequently presented in Section 3. Section 4 presents the conclusions from the experimental work.

## 2. Detection Concept and Simulation

As explained earlier, CLAS refers to antennas sandwiched between multiple layers of composite. A schematic depicting a multilayer composite with a patch antenna fabricated on one of the layers is shown in Figure 1. As can be seen from the figure, a patch antenna operating in the C-band is sandwiched between a paper-based honeycomb substrate and a GFRP sheet from above. The planar resonator probe (UHF probe) is used for inspecting the antenna embedded inside (patch in this case) and placed at a standoff distance *d* from the surface of the top cover. The probe radiates the structure under test, such that it excites current on the antenna surface underneath it. The UHF probe is very sensitive to current distributions and any deformity that represents a structural anomaly can easily be detected. Moreover, a larger structure can be interrogated by the probe owing to its greater wavelength, avoiding the possibility of false detection. This is highly likely for probes operating at a higher frequency and shorter wavelength, where the null in the electric field can easily be mistaken for structural deformity in the SUT.

The potential utility of microwaves in the non-destructive evaluation of composites is well known. However, the selection of the microwave probe for the detection of defects in a given application is critical. While near-field aperture probes have been demonstrated to be effective for inspecting composites, the distinct advantage of using a planar resonator probe, or a UHF probe, is explained herein. The planar resonator probe introduced in [30] is highly sensitive to discontinuities in current distribution and operates at a low frequency of 300–900 MHz.

For simulation studies and analysis, a representative CLAS based on the structure shown in Figure 1 was designed and simulated using CST Studio Suite. The antenna was sized appropriately to operate in the C-band (4–8 GHz), which is commonly used in practical applications. A copper-grounded square patch of dimensions 27 mm by 25.88 mm was created on an air substrate with a dielectric value similar to a paper-based honeycomb substrate. A 15 mm long and 4 mm wide quarter wavelength transformer, in addition to a feed measuring 15 mm in length and 14.75 mm in width, was also made. The width of the feed was selected to ensure 50-ohm matching. Finally, a 1.5 mm thick substrate of GFRP was used to cover the antenna, as depicted in Figure 1. The structure was simulated as built and is referred to as “healthy antenna”. Subsequently, a 2 mm wide cut was introduced in the transformer line, in addition to a diagonal slit on the patch itself to emulate a “damaged antenna”. The reflection coefficient of the healthy and damaged antenna obtained from simulation is presented in Figure 2. It is evident from the figure that the antenna resonates sharply at the intended frequency band ~4.6 GHz with a narrowband response expected from a simple square microstrip patch antenna. However, the damaged antenna renders a less efficient response and barely resonates at −10 dB. To ensure and further investigate the operability of the damaged antenna, the radiation pattern and electric field distributions were plotted.

Figure 3a,b show the simulated antenna gain radiation patterns in the two principal planes (E, H) for the healthy and damaged scenarios, respectively. The farfield radiation pattern is another important parameter that is used to characterize an antenna. As can be seen in Figure 3a, the simulated healthy microstrip patch antenna radiates unidirectionally towards the broadside as expected, with minimal back lobes and a symmetric pattern in the E-plane. While the damaged version has similar characteristics, the maximum gain is significantly reduced (−5.8 dBi). Figure 3b shows the orthogonal polar cut (H-Plane) of the gain pattern for both antenna states (healthy and damaged) as annotated in the figure. It can be seen that the patterns are similar. However, the damaged pattern directivity is reduced in the broadside relative to the healthy state while maximizing at 400 in both directions suggesting the main beam was split. The antenna radiation pattern shows that the antenna is operational in both states, albeit with less gain due to greater losses. The total efficiency of both antenna states is reported to be 93% and 84%, respectively.

The linear 3-D radiation pattern shown in Figure 4a,b and the electric field distribution along the patch antenna depicted in Figure 4c,d further corroborate the fact that the damaged antenna is still radiating. In Figure 4c, it can be observed that the electric field density from the patch is high, indicating a healthy antenna, and is in line with expectations. For the damaged patch antenna however, the cut on the transformer creates a radiating stub as seen in Figure 4d with a relatively denser electric field when compared to the patch. While the electric field intensity is lower around the patch, it can still be deemed radiating. This explains the antenna’s resonance at a higher frequency, as presented earlier in Figure 2. Consequently, the post-damage functionality of the antenna is established based on these observations.

After confirming the operability of the embedded antenna, simulations were performed to analyze the sensitivity and performance of the imaging probes. To this end, the UHF probe and a K-band rectangular aperture were modeled in CST microwave studio. The CLAS model shown in Figure 1 was used as structure under test (SUT). A picture of the simulation setup to assess the probe response is depicted in Figure 5. A 0.25 mm wide slit was made onto the patch creating a healthy and damaged version of the CLAS, as before. The reflection coefficient of the probes was measured for both healthy and damaged cases and the change in magnitude of the reflection coefficient (linear) was recorded for both probes and is tabulated in Table 1. It is evident that both probes detected a change in the reflection coefficient when placed over the damaged patch covered under the GFRP sheet. The UHF probe rendered a relatively higher change (more than 2.5 times) when compared to the rectangular aperture probe. Additionally, the probe is also polarization-independent and can detect slits/cuts regardless of their orientation on the embedded antenna. Based on the aforementioned observations, it is evident from the simulation that the UHF probe is relatively more sensitive and better suited for such applications.

## 3. Measurement Results

To demonstrate the utility of microwaves in inspecting antennas embedded in composites, a representative sample was made. A photograph of the various layers constituting the composite structure is shown in Figure 6a. The lowermost layer is a 0.1 mm thick copper foil, which also serves as the ground plane for the antenna. The middle layer is an aramid paper-based honeycomb substrate of thickness 3 mm. To ensure the smooth laying of the antenna, the honeycomb substrate was covered with a 0.15 mm thick paper tape. A simple patch antenna operating in the C-band was then fabricated and sandwiched between the honeycomb substrate and a GFRP sheet. A rectangular patch antenna was chosen for the demonstration, since it is the most common in such applications. Initially, the patch was made out of copper foil, as depicted in Figure 6b.

The patch antenna dimensions measured 27 mm by 25.88 mm, with a quarter-wavelength transformer width of 4 mm and length of 15 mm, in line with the simulations. The 50-ohm feed of the antenna is also 15 mm long with a width of 14.75 mm to ensure 50-ohm matching. An SMA connector was then soldered onto the patch and connected simultaneously with the ground plane. Finally, a 1.5 mm thick GFRP sheet was used to cover the antenna from the top, as illustrated in Figure 6c. Subsequently, to emulate a damaged antenna, a 2 mm wide cut was deliberately introduced in the transformer line. Additionally, a diagonal cut (slit) was also made on the patch itself, as annotated in Figure 6b. The antenna was then characterized pre- and post-damage using a Keysight N5225A performance network analyzer (PNA). The measured antenna response, in terms of the reflection coefficient, was plotted against the simulated response and is shown in Figure 7. It is evident that a close match between simulation and measurement was obtained. The minor differences in the response, particularly the reflection coefficient of measurement not at zero, could be explained due to the fact that calibration was performed only up to the cable and not up to the SMA connector. The slight shift in resonance frequency of the damaged antenna could be attributed to fabrication tolerances and human error while introducing the damage into the patch antenna. In line with the simulation results, the antenna was radiating in the post-damage state. After ensuring that the antenna was working even after being damaged, the structure was then inspected using the planar resonator probe to non-destructively test and portray any damage/irregularities in the structure through high resolution images.

From an NDT point of view, the detection of such minor damage in the antenna, which does not hinder its performance significantly, is crucial. It is indeed very difficult to determine the extent of damage merely from the reflection coefficient response of the antenna. Unless detected during an early stage, this may lead to the complete failure of antennas embedded in composites, and consequently the smart structure. To this end, the GFRP-covered composite with an embedded antenna was imaged using the UHF probe. The PNA mentioned before (Keysight N5225A) was used as a source/detector. The microwave imaging setup employed for all imaging experiments is shown in Figure 8. For all imaging experiments, the UHF probe was fixed at a liftoff of 1 mm from the topmost surface of the composite sample. The probe was held stationary, while the composite sample was moved in controlled but predetermined step sizes using an automated scanning table equipped with motorized linear stages. The sample was moved with a step size of 1 mm along each axis. During each step, the complex reflection coefficient (S_11_) was recorded in and saved as a two-dimensional matrix. The scan data thus saved were then processed in Matlab software (R2022 a version) by indexing the complex data for all frequency swept points over the total scan area. Subsequently, the complex data for a particular frequency point (of interest) were then selected and a magnitude image was created by plotting the absolute value of the complex data over the scan area. Similarly, the angle of the complex data was plotted over the scan area to obtain the phase image.

Two different scans were performed for the healthy and the intentionally damaged antenna-embedded composite. The images of both samples generated by the UHF probe are shown in Figure 9. The magnitude and phase images are placed side by side in each case. The images of the healthy antenna embedded in the composite are depicted in Figure 9a,b while the images for the damaged antenna-embedded composite (magnitude and phase) are presented in Figure 9c,d. The UHF probe produced a sharp depiction of the patch antenna structure with clear distinction from the background. The transformer line and the 50-ohm feed are also clearly visible in the image. On the other hand, magnitude and phase images of the damaged antenna also portray the patch antenna outline sharply. Moreover, the 2 mm wide cut and the diagonal slit in the patch are conspicuous in the image. The gap created in the transformer line due to the cut manifests itself in both magnitude and phase images. Furthermore, the diagonal slit on the patch is distinguishable from the intact area with clear demarcation, particularly in the magnitude image.

To further investigate microwave NDT for antenna-embedded composite structures, a patch antenna operating in the C-band was fabricated using a highly conductive copper mesh with a surface resistance <0.03-ohm. The same mesh was utilized as the ground plane for the antenna. It is to be noted that in actual applications the antenna will be meshed. Microwave NDT methods have been used extensively to detect cuts in sheets, but not metallic meshes. Therefore, the imaging performance was initially demonstrated on the copper foil to establish a reference to something common (sheet/foil) before introducing the mesh application. Similarly to the previous experimental procedure, two scenarios representing a healthy and damaged antenna were considered. For each imaging scan, the antenna was covered with the 1.5 mm thick GFRP sheet. A higher magnification picture of the healthy, damaged and GFRP-covered antenna is shown in Figure 10a–c respectively. The UHF probe employed in the previous experiment was used to image the two sets of SUT.

The magnitude and phase images thus produced by the probe for the healthy antenna are shown in Figure 11a,b, respectively. As evident from the images, the probe produced very sharp depictions of the patch, the transformer and the feed. The background in the image is also very distinct and clear, particularly in the magnitude response. Subsequently, the mesh was then intentionally damaged by severing the transformer line with a 2 mm wide cut, and a diagonal slit was made across the patch, as illustrated in Figure 10b. The damaged patch antenna-embedded composite structure was then imaged using the UHF probe. The magnitude and phase images (c.f. Figure 11c,d) depict the induced damage very distinctly. The cut in the transformer line is apparent in the magnitude image. Furthermore, the diagonal slit on the patch is also faithfully rendered in both magnitude and phase images. In order to benchmark the imaging capability of the UHF probe, the damaged patch antenna was imaged by a standard K-band rectangular aperture probe. The purpose of benchmarking the imaging results to a K-band rectangular aperture is because it is widely used in microwave NDT applications. It is noted here that the standard rectangular waveguide aperture probe is linearly polarized, while the UHF probe is not (it is polarization-independent). For this imaging experiment, the K-band rectangular aperture was positioned such that its polarization was along the direction of the patch antenna feed. The magnitude and phase images obtained from the rectangular aperture probe are depicted in Figure 11e,f. While the cut on the transformer line is visible in the magnitude image, the diagonal slit is not discernible. The target is almost missed by the rectangular aperture probe.

This experiment highlights the sensitivity of the UHF probe to the presence of thin cuts/slits, as it is particularly sensitive to changes in current distributions. To further evaluate the imaging performance of the probe, the signal-to-noise ratio (SNR) values of the images produced by the UHF probe and the K-band rectangular aperture probe were calculated. Table 2 shows the SNR values of the images, and it can de deduced that the UHF probe image has a significantly high (more than 8 times) SNR in contrast to the K-band rectangular aperture probe image. The qualitative as well as quantitative metrics therefore highlight the proficiency of the UHF probe in terms of the NDT of the antenna-embedded composites. It is further remarked that the lateral spatial resolution of the UHF probe has been established to be 3 mm [30], while that of the K-band rectangular aperture probe is limited to half the broad dimension, around 5.35 mm in this case. Additionally, using high- frequency probes for inspection in this application may not necessarily yield the best results, since the antenna under test (patch in this case) might be a good portion of the operating wavelength. Hence, by employing a low-frequency probe, such as the UHF probe (used herein), the structure under inspection is much smaller than the wavelength and thus better imaging results are obtained.

To summarize, the planar resonator probe utilized herein offers significant advantages over the conventional rectangular aperture probe. Firstly, it is polarization-independent and can detect cuts/slits regardless of their orientation, which is not the case for rectangular apertures. Secondly, it is highly sensitive to current distributions and operates at a low frequency thereby having a larger wavelength. This feature is considered to be favorable in these applications, since it limits false detections, as explained earlier. Thirdly, the probe renders high-resolution images with relatively better SNR at such low frequency that are comparable to images generated by probes operating at much higher frequencies. Overall, the effectiveness of microwave NDT as a modality for the in situ inspection of composite smart structures is shown in this work.

## 4. Conclusions

In this work, the utility of microwave NDT for the structural health monitoring of antenna-embedded composite structures, or smart structures, has been shown. A low-frequency planar resonator probe was employed to generate high-resolution images of the structure under test. The typical advantages of the resonator probe, including high sensitivity to current distributions and low frequency of operation, were deemed favorable for the non-destructive evaluation of antennas embedded in composites. This fact was verified from simulation study and through experimental trials wherein a C-band patch antenna fabricated on a honeycomb substrate sandwiched between a GFRP sheet (top) and a metallic mesh (bottom) was considered. The antenna was constructed from copper foil and, subsequently, from a highly conductive copper mesh. The antenna was intentionally damaged by introducing narrow slits in the patch and a cut on the transformer line, creating two scenarios representing a healthy and a damaged antenna. The aforementioned defects were successfully detected by the planar resonator when imaged. The defects were depicted clearly, with high contrast and distinction from the background. Furthermore, to highlight the effectiveness of the resonator probe, the damaged mesh antenna was imaged using a standard K-band rectangular aperture probe. Not only were the resonator probe images qualitatively better when contrasted to the rectangular aperture probe images, but they also had a relatively very high SNR (>8 times). While the efficacy of the resonator probe was elucidated herein, the significance of the NDT of such composite smart structures was the primary focus of this work. More importantly, the potential of microwave NDT as an inspection tool for the evaluation of CLAS was demonstrated herein.

## Figures and Tables

**Figure 1 sensors-23-03200-f001:**
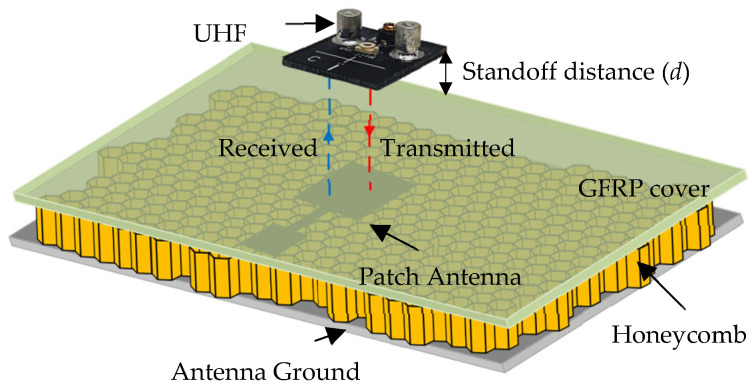
A schematic showing the antenna embedded in a composite structure, and a UHF probe used for inspection of the structure.

**Figure 2 sensors-23-03200-f002:**
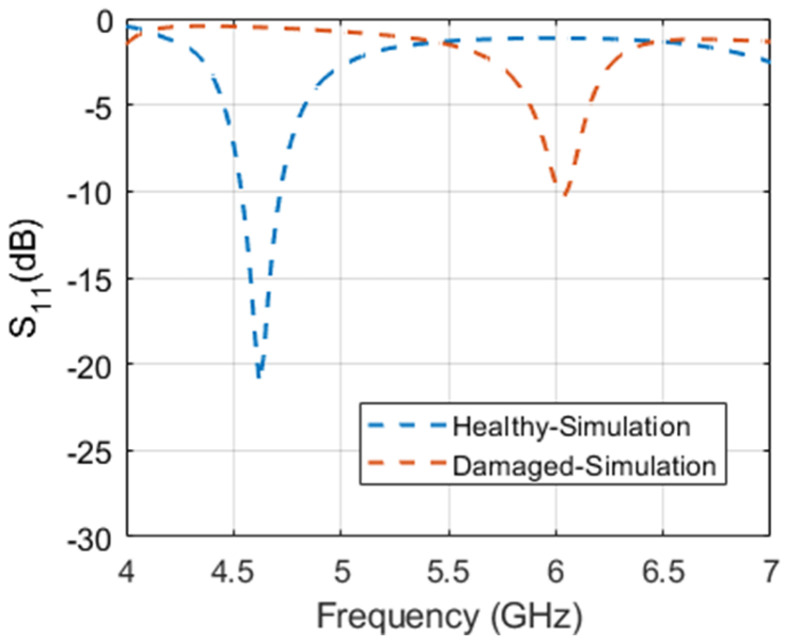
Magnitude of the reflection coefficient of the simulated patch antenna in pre- and post-damage infliction.

**Figure 3 sensors-23-03200-f003:**
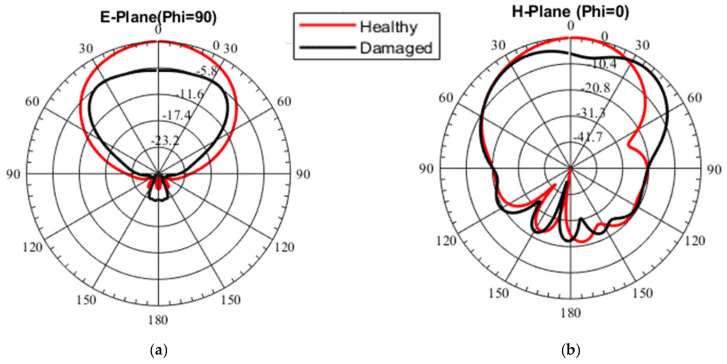
Simulated gain radiation patterns of healthy and damaged patch antenna in the (**a**) E-plane and (**b**) H-plane.

**Figure 4 sensors-23-03200-f004:**
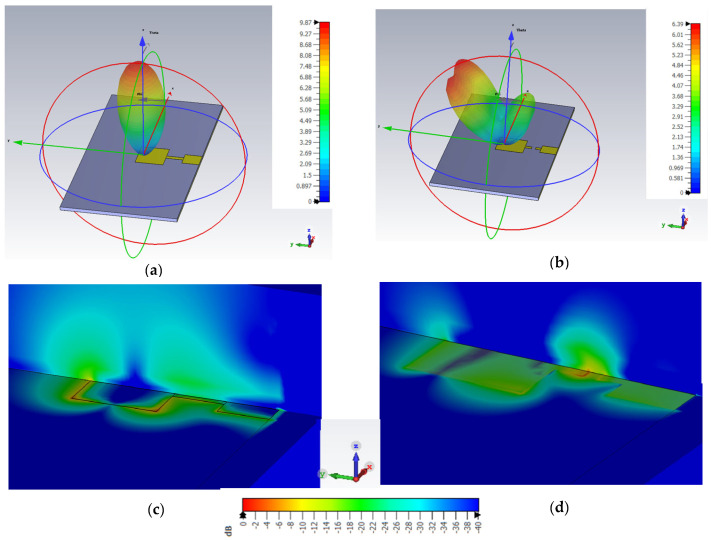
The 3-D linear radiation pattern for (**a**) healthy and (**b**) damaged antenna and electric field distributions of the (**c**) healthy and (**d**) damaged patch antenna.

**Figure 5 sensors-23-03200-f005:**
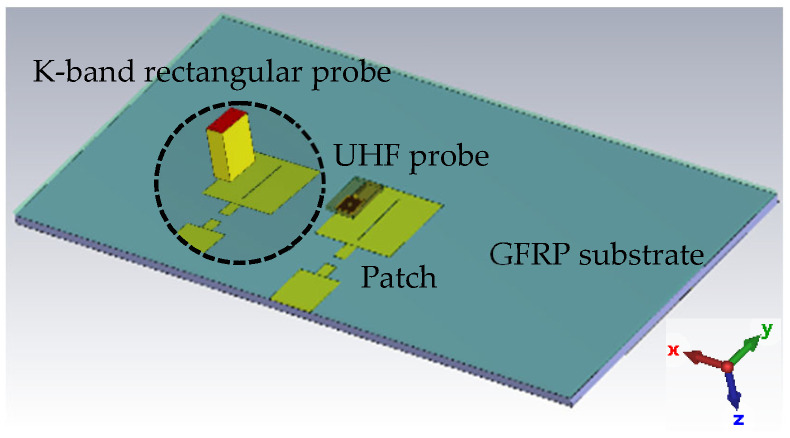
Picture of the simulation setup to assess probe sensitivity for damaged patch antenna.

**Figure 6 sensors-23-03200-f006:**
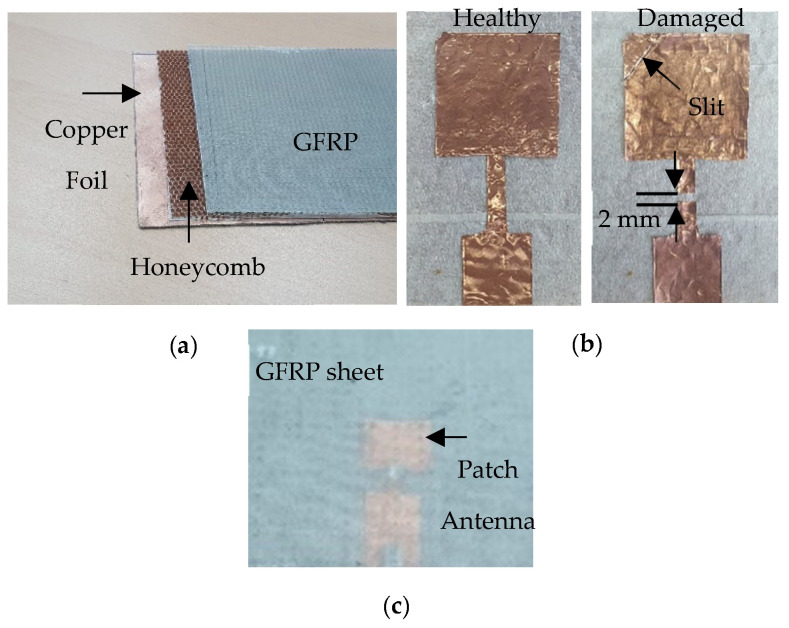
(**a**) Picture of the individual layers of the composite structure, (**b**) the C-band patch antenna fabricated on a honeycomb substrate and (**c**) the embedded antenna covered with a GFRP sheet.

**Figure 7 sensors-23-03200-f007:**
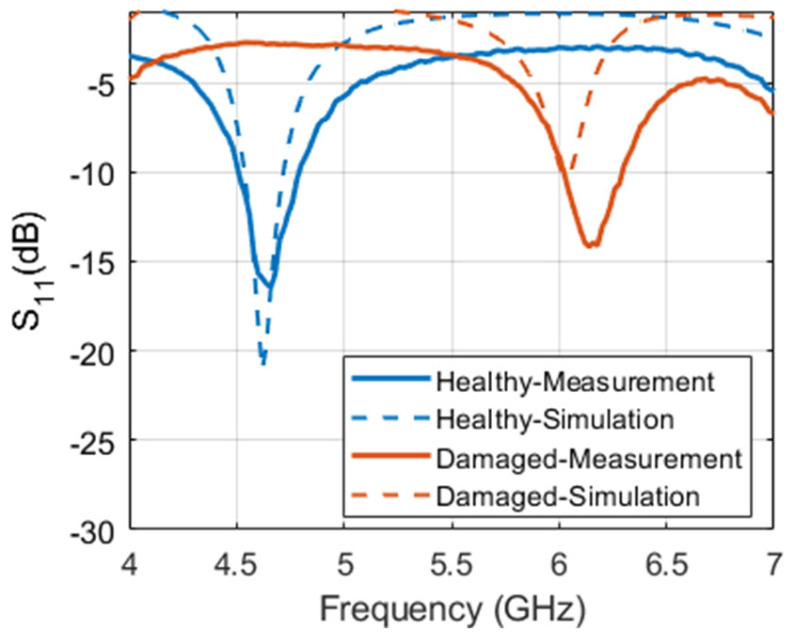
Measured magnitude of reflection coefficient of the patch antenna before and after inflicting damage.

**Figure 8 sensors-23-03200-f008:**
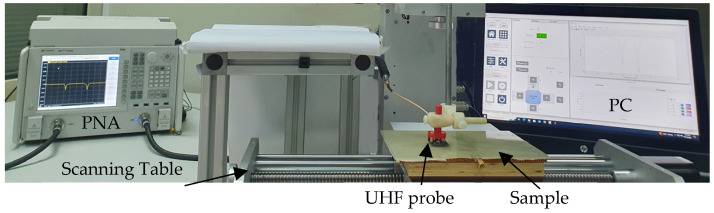
Photograph of the imaging setup.

**Figure 9 sensors-23-03200-f009:**
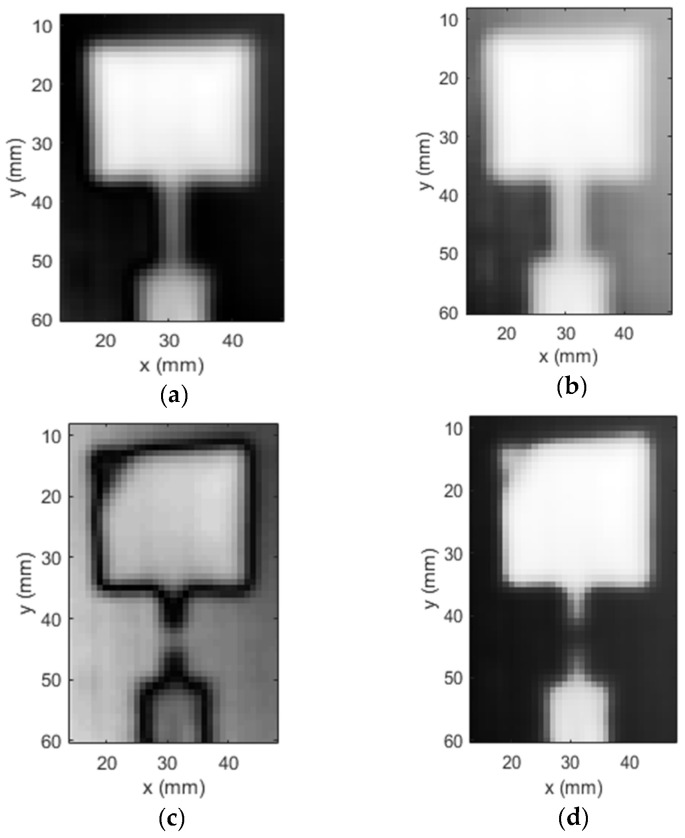
Magnitude and phase images of the (**a**,**b**) healthy and (**c**,**d**) damaged C-band patch antenna fabricated using copper foil, produced by the UHF probe.

**Figure 10 sensors-23-03200-f010:**
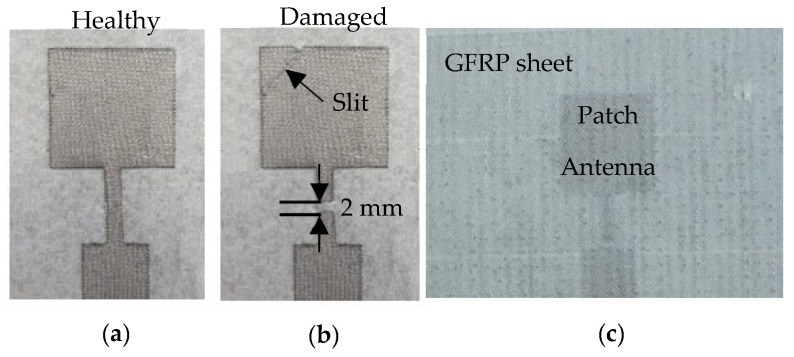
Photographs of (**a**) healthy and (**b**) damaged patch antenna fabricated using metal mesh, and (**c**) damaged antenna covered with a GFRP sheet.

**Figure 11 sensors-23-03200-f011:**
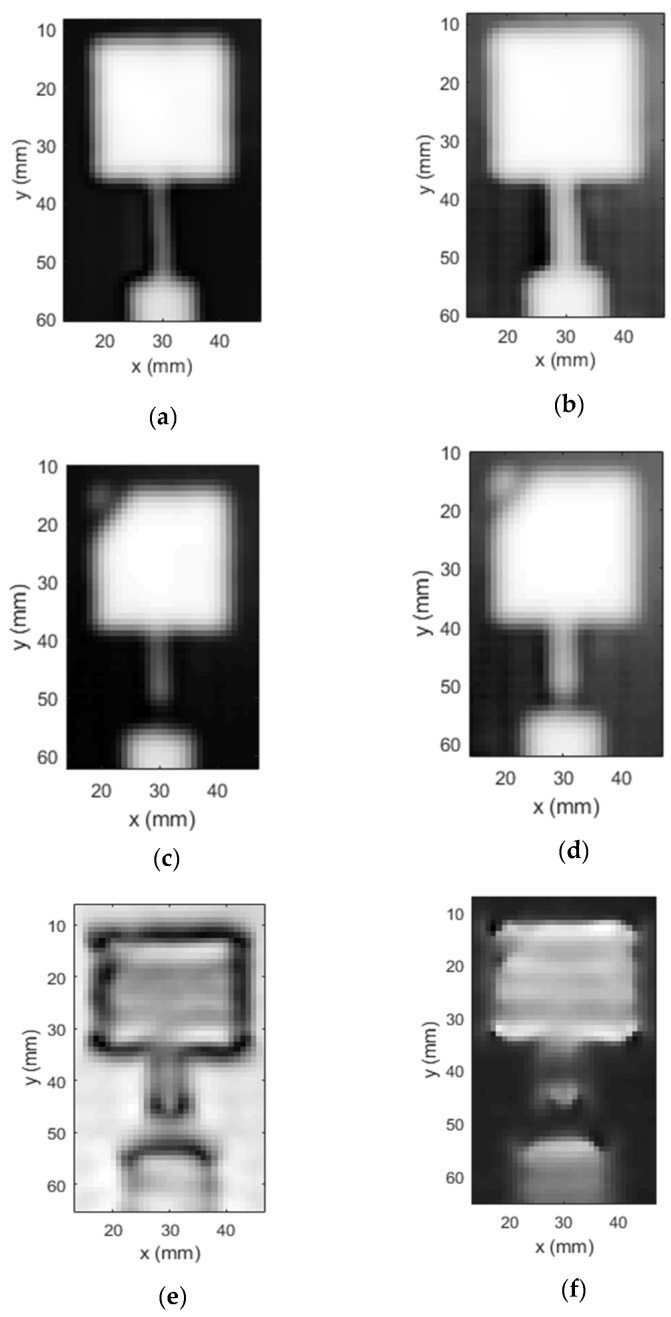
Magnitude and phase images of the (**a**,**b**) healthy and (**c**,**d**) damaged C-band patch antenna fabricated using metal mesh, produced by the UHF probe, and (**e**,**f**) damaged patch produced by K-band rectangular aperture probe.

**Table 1 sensors-23-03200-t001:** Change in magnitude of reflection coefficient of imaging probes.

Probe Type	∆|S_11_| (Linear)
UHF Probe (at 510 MHz)	0.72
K-band rectangular aperture probe (at 25 GHz)	0.27

**Table 2 sensors-23-03200-t002:** The SNR values of damaged patch antenna (mesh) images.

Probe Type	SNR Value
UHF Probe	541.17
K-band rectangular aperture probe	64.52

## Data Availability

Data available on request due to restrictions.

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
