# Peer review of "Microwave NDT of Smart Composite Structures with Embedded Antennas"

_sensors, 2023, doi:10.3390/s23063200_

Round 1

Reviewer 1 Report

Authors studied the detection of defects in the antenna embedded composite structures using a planar resonator probe operating in the UHF frequency range. This topic is not original but relevant to the field of microwave NDT inspection of composite structures and deserve consideration. They concluded that the potential of microwave NDT evaluation of load-bearing antenna structures was demonstrated. The manuscript corresponds to the Sensors. The Introduction is quite complete but a little bit too long. The methodology of the study is described in sufficient detail. The text is reasonably clear and easy to read. All structural units of the manuscript are logically interconnected. The manuscript contains important scientific results for practice. 

 Comments and suggestions:

  1. The Introduction is recommended to make shorter.
  2. It is desirable that the lateral spatial resolution of the inspection technique be indicated in the text.
  3. The list of references contains at least of 14 points of self-citation (i.e. 40%). Evidently, self-citation has to be reduced.
  4. The background in Fig.7 and Fig.9 should be diminished while the images of the antennas enlarged.
  5. The repeated explanation of GFRP abbreviation in line 203 is excessive.

Reviewer 2 Report

The article proposed and demonstrated the use of planar resonator probe to to detect antenna damages. Meaningful results and insights were presented in the article. I have following comments.

1. Section 2 "Detection Concept and Simulation" seems to be less relevant. It mainly demonstrates the s11/radiation pattern/current distribution differences between damaged antenna and healthy antenna, but no simulation study result was conducted in this section regarding probing and imaging. As simulation study is an important tool to understand the problem, I suggest to improve this section. It may provide additional insights as well.

2. In Section 3, can you give more details of how the measurements were conducted? Pictures of your measurement equipment, setups, probes would be helpful. Also, can you add description about how the images were reconstructed from raw measurement results?

3. It would be helpful to emphasize the main contribution/novelty of the article. Also, it would also be good to dig into how/why planar resonator probe is a good fit for this application from simulation or experiment raw data.

Round 2

Reviewer 1 Report

After revision the paper can be published.

Author Response

Point 1: After revision the paper can be published.

Response 1: Thank you very much.

Reviewer 2 Report

Thanks for the revision! It would be good to improve resolutions of some figures as some legends are hard to read.

Author Response

Point 1: Thanks for the revision! It would be good to improve resolutions of some figures as some legends are hard to read.

Response 1: Thank you for the comments and suggestion. The legends of Figures 3, 4 and 8 have now been made clear in the revised manuscript.